# Antagonistic Trends Between Binding Affinity and Drug-Likeness in SARS-CoV-2 Mpro Inhibitors Revealed by Machine Learning

**DOI:** 10.3390/v17070935

**Published:** 2025-06-30

**Authors:** Anacleto Silva de Souza, Vitor Martins de Freitas Amorim, Eduardo Pereira Soares, Robson Francisco de Souza, Cristiane Rodrigues Guzzo

**Affiliations:** Department of Microbiology, Institute of Biomedical Sciences, University of São Paulo, Sao Paulo 05508-000, Brazil; vitormartins@usp.br (V.M.d.F.A.); eduardo_soares@usp.br (E.P.S.); rfsouza@usp.br (R.F.d.S.); crisguzzo@usp.br (C.R.G.)

**Keywords:** drug discovery, hit-to-lead optimization challenges, main protease, SARS-CoV-2, machine learning, molecular dynamics simulations

## Abstract

The SARS-CoV-2 main protease (Mpro) is a validated therapeutic target for inhibiting viral replication. Few compounds have advanced clinically, underscoring the difficulty in optimizing both target affinity and drug-like properties. To address this challenge, we integrated machine learning (ML), molecular docking, and molecular dynamics (MD) simulations to investigate the balance between pharmacodynamic (PD) and pharmacokinetic (PK) properties in Mpro inhibitor design. We developed ML models to classify Mpro inhibitors based on experimental IC_50_ data, combining molecular descriptors with structural insights from MD simulations. Our Support Vector Machine (SVM) model achieved strong performance (training accuracy = 0.84, ROC AUC = 0.91; test accuracy = 0.79, ROC AUC = 0.86), while our Logistic Regression model (training accuracy = 0.78, ROC AUC = 0.85; test accuracy = 0.76, ROC AUC = 0.83). Notably, PK descriptors often exhibited opposing trends to binding affinity: hydrophilic features enhanced binding affinity but compromised PK properties, whereas hydrogen bonding, hydrophobic, and π–π interactions in Mpro subsites S2 and S3/S4 are fundamental for binding affinity. Our findings highlight the need for a balanced approach in Mpro inhibitor design, strategically targeting these subsites may balance PD and PK properties. For the first time, we demonstrate antagonistic trends between pharmacokinetic (PK) and pharmacodynamic (PD) features through the integrated application of ML/MD. This study provides a computational framework for rational Mpro inhibitors, combining ML and MD to investigate the complex interplay between enzyme inhibition and drug likeness. These insights may guide the hit-to-lead optimization of the novel next-generation Mpro inhibitors of SARS-CoV-2 with preclinical and clinical potential.

## 1. Introduction

Despite efforts in vaccination in several countries, more than 777 million cases and 7.05 million deaths by Severe Acute Respiratory Syndrome Coronavirus 2 (SARS-CoV-2) have been reported since the beginning of the Coronavirus disease 2019 (COVID-19) pandemic until 14 May 2025 [1]. Several SARS-CoV-2 variants of concern (VOC) such as the Alpha (B.1.1.7), Beta (B.1.351), Gamma (P.1), Delta (B.1.617.2), and Omicron (B.1.1.529) and its sublineages emerged, all showing higher transmission rates and immune system evasion than the Wuhan strain (wild type, WT) [2,3]. In particular, the Omicron variant emerged at the end of November 2021, and within three months, it rapidly prevailed over the Delta variant [2,3]. The lineage B.1.529 (Clade 21M) gave rise to its sublineages, BA.1 (Clade 21K), BA.2 (Clade 21L), BA.2.12.1 (Clade 22C), BA.4 (Clade 22A), and BA.5 (Clade 22B). As of 14 May 2025, the proportion of Omicron variants summed almost 100%, in which the LP.8.1 (Clade 25A) represented more than ~34% followed by XEC sub-variant (Clade 24F) with 24%, NB.1.8.1 (Clade 25B) with 16%, and KP.3.1.1 with 12% [3]. Recently, the discovery of the HKU5-CoV lineage 2 (HKU5-CoV-2) in bats has raised concerns, as this virus efficiently binds human ACE2 (hACE2) and exhibits broad host tropism [4]. Comparative genomic analyses reveal significant sequence divergence in the Spike protein-coding region relative to other Coronaviruses, underscoring the potential for novel cross-species transmission [4].

At the molecular level, after viral infection, viral mRNA is released into the cytosol of the host cell and translated into two polyproteins, pp1a and pp1ab, which compose non-structural proteins responsible for different functions in the viral infection process [5]. These polyproteins are cleaved by two proteases, PLpro (papain-like protease), responsible for cleaving three sites in the pp1a and pp1ab, and 3CLpro (3-chymotrypsin-like protease), also known as Mpro (main protease), that cleaves at least eleven cleavage sites [5,6]. The Mpro is a cysteine protease and plays a key role in the viral infection process. Its active form is a homodimer, in which each protomer consists of three domains, known as domains I, II, and III. In a previous study, we suggested that residues His41–Cys145–Asp187 compose the catalytic triad of Mpro, being essential for the catalytic process [5]. Cysteine is a conserved element within the Coronaviridae family, functioning as a nucleophile crucial for the proteolytic process [5,7]. The 3CLpro protein’s active site can be divided into the subsites S1, S1′, S2, and S3/S4, adhering to the Schecter–Berger nomenclature for proteases [5,8]. This arrangement underscores the critical contribution of specific residues to enzymatic activity in the context of viral infection.

To date, more than 55,429 chemical structures have been experimentally evaluated against Mpro and reported in the scientific literature [9]. The extensive diversity in chemical compounds arises from the large active site with distinct subsites, making them highly potent against the main enzyme of the novel Coronavirus. Although this diversity may lead to the existence of a broad range of bioactive compounds, as shown in Figure 1, it is not clear why only a few 3CLpro antiviral compounds are available for clinical use. Additionally, previous studies show that 3CLpro inhibitors of MERS-CoV, SARS-CoV, and SARS-CoV-2 have different binding affinities [10], thus raising the possibility that 3CLpro inhibitors could be less effective in VOCs [5,11].

A substantial number of chemical structures have been screened against SARS-CoV-2 Mpro [9], and only a small fraction of the compounds have advanced from preclinical to clinical stages [5]. In our recent study, we explored pharmacokinetic (PK) challenges that may significantly influence the efficacy of these compounds in preclinical and clinical phases [5]. One example is Boceprevir, a hepatitis C virus protease inhibitor, which demonstrated promising affinity for Mpro and antiviral activity against SARS-CoV-2 (IC_50_ = 4.1 μM and EC_50_ = 1.3 μM) [5,12]. In the case of hepatitis, this antiviral is only available by controlled prescription. In the case of possible use in SARS-CoV-2, further studies are required to advance this inhibitor to clinical trials due to PK limitations, including human cytochrome P450 isoform (CYP3A4) inhibition [5,12]. To address similar PK challenges, the combination of Nirmatrelvir with Ritonavir—marketed as Paxlovid—was developed, utilizing Ritonavir’s CYP3A4 affinity and plasma protein binding properties to enhance therapeutic efficacy [5]. Despite these advances, a lack of predictive models to guide the optimization of PK properties continues to limit the search for novel Mpro inhibitors. To address this gap, we investigated the interplay between PK and pharmacodynamic (PD) properties using machine learning (ML) and molecular dynamics (MD) simulations. We developed classification models employing different ML algorithms such as Support Vector Machines (SVM) and Logistic Regression (LR). These models were trained using a database of Mpro inhibitors using different biophysics analysis that includes the value of IC_50_ from FRET assays (353 compounds), fluorescence assays (1595 compounds), and mixed datasets (FRET, fluorescence, and SPR assays) (1943 compounds). Our analysis revealed that hydrophilic features of inhibitors enhance Mpro binding but compromise PK properties. We also simulated different potent Mpro inhibitors and determined the free-energy landscape (FEL) by MD trajectories and principal component analysis. Our MD analysis also showed that hydrogen bonding, hydrophobic, and π–π interactions in the subsites S2 and S3/S4 are fundamental for binding affinity. These findings highlight the importance of targeting the subsites S2 and S3/S4 to balance PD and PK properties and provide a computational framework for future rational design of Mpro inhibitors, offering a pathway for hit-to-lead optimization.

## 2. Materials and Methods

### 2.1. Dataset Construction and Data Preprocessing

We identified 18 articles on SARS-CoV-2 Mpro inhibitors, of which 54,087 molecular entries with biological activity information were provided in the AI-driven Structure-enabled Antiviral Platform (ASAP) [9,13]. For the remaining 17 papers, data extraction was performed manually and curated using Open Babel [14,15] and MarvinSketch [16]. These efforts resulted in an initial, uncleaned dataset comprising 55,419 molecules. Data transformations included conversion to numerical format and log-scale adjustment (pIC_50_ = −log IC_50_) in order to facilitate statistical analyses. The dataset was organized and analyzed to investigate the correlation between different IC_50_ measurement methods, specifically focusing on experimental assays involving FRET, fluorescence, and SPR as measurement techniques. The dataset was imported using the pandas library, and unnecessary columns were dropped. We developed a set of scripts that provide methods for calculating substrate affinity and mean binding constants, treating missing or censored IC_50_ values. The predicted *K*_M_ calculation estimates the Michaelis-Menten constant (*K*_M_) based on substrate and protein concentrations. Based on our published dataset in our previous study [17], we defined that if the protein concentration ([Mpro]) exceeds 50 nM and the substrate concentration ([subst]) is greater than 15 µM, *K*_M_ was set to 24 µM. If only the substrate concentration is above 15 µM, KM was assigned a value of 75.41 µM; otherwise, it defaults to 40.87 µM (mean of these data). The mean calculation of *K*_i_ and *K*_D_ values assumes that both values present a similar magnitude, calculating the mean when both values are available. Indeed, in our dataset, we observed that both have similar magnitude (dataset is available in Appendix A). If only one of the values was present, that value was used directly. If neither *K*_i_ nor *K*_D_ was available, the result was NaN. This approach ensures a representative binding constant even when experimental data is incomplete. We processed IC_50_ values, treating censored data and cleaning invalid entries. We retained values starting with ‘>’ or ‘<’ and recognized both integers and decimals. If the value did not meet these criteria, we attributed NaN. We ensured data integrity, preserving information from censored measurements while filtering out invalid data points. Thus, we established a reliable framework for processing enzymatic inhibition data, facilitating more accurate quantitative modeling of compound potency.

We computed the pIC_50_ value from experimental IC_50_ or *K*_i_/*K*_D_ measurements, accounting for adjustments based on the concentration of the target protein (Mpro) and the Michaelis-Menten constant (*K*_M_). The input data consisted of IC_50_ values converted to nanomolar, the concentration of Mpro in nanomolar, and the predicted *K*_M_, *K*_i,_ or *K*_D_ value (or mean of both). Initially, the method attempted a direct conversion of IC_50_ to pIC_50_ when the IC_50_ value was numerical, positive, and below 90 nM, using the equation as follows:$${pIC}_{50}=-{log}_{10}\;\left(\frac{{IC}_{50}}{10^{6}}\right)$$

If the direct conversion was not possible, an adjustment factor was calculated considering the expression as follows:$$factor=1+\left(\frac{\frac{\left[M_{pro}\right]}{10^{9}}}{\frac{K_{M}}{10^{6}}}\right)+\left(\frac{\frac{\left[M_{pro}\right]}{10^{9}}}{\frac{2}{1}}\right)$$

For IC_50_ values with ‘>’ or ‘<’, the method computed pIC_50_ using the adjusted *K*_i_/*K*_D_ value was as follows:$${pIC}_{50}=-{log}_{10}\;\left(\frac{\frac{K_{i}}{K_{D}}\times factor}{10^{6}}\right)$$

This approach enabled the estimation of pIC_50_ even when the IC_50_ value was censored or ambiguous. In cases where IC_50_ was missing or invalid but *K*_i_/*K*_D_ was a positive numerical value, pIC_50_ was calculated using the same adjustment factor and equation. Our strategy facilitated implementation by handling exceptions such as non-numeric or invalid types, preventing interruptions during data processing. This approach provided a robust conversion of enzymatic inhibition data to pIC_50_, integrating adjustments for concentration and affinity constants, thereby supporting quantitative analysis of inhibitor activity even with incomplete or censored measurements. For pIC_50_ values below 5, molecules were considered inactive against Mpro; otherwise, they were classified as active against this enzyme. All outputs were systematically organized and made easily accessible for further analysis and visualization. The resulting datasets included Dataset 1: FRET (353 molecules); Dataset 2: fluorescence (1595 molecules); and Dataset 3: Combined dataset (FRET, SPR, and fluorescence; 1943 molecules). Dataset 3 was specifically structured to evaluate whether assay types independently influenced machine learning predictions, regardless of inhibition assay types.

### 2.2. Computational Environment

The computational analyses were performed in Python 3.12.2, utilizing a suite of scientific libraries including NumPy [18], Pandas [19], RDKit [20], scikit-learn [21], Plotly [22], Matplotlib [23,24], and seaborn [25]. These tools support data manipulation, molecular descriptor computation, dimensionality reduction, and data visualization. Protonation states of molecular structures were predicted using Dimorphite-DL [26], which we adapted to enable parallel processing across datasets and to constrain the analysis within a physiologically relevant pH range (6.8 to 7.2), allowing a maximum of one variant per molecule. We computed 2D molecular descriptors using RDKit, where molecules were converted from SMILES strings to RDKit Mol objects, followed by molecular descriptor calculations. We also generated 3D descriptors by embedding molecules with the ETKDG algorithm, implemented on RDKit, where hydrogen atoms were added, and 3D conformers were generated. Due to the high dimensionality inherent in molecular descriptor datasets—known to complicate machine learning (ML) applications by increasing overfitting risk, computational cost, and reducing model interpretability [27]—we applied dimensionality reduction based on Pearson’s correlation coefficient, removing highly correlated features. Features with missing values or zero variance were excluded, and remaining missing values were imputed with mean values. Further procedural details are provided at https://github.com/anacletosouza/Mpro_AI-ML_pipeline/ (accessed on 11 June 2025) and Appendix A. Feature sets derived from various filtering strategies (see Appendix A) were used to train predictive models, whose performance was evaluated using precision, recall, F1-score, and accuracy as follows:$$Precision=\frac{TP}{TP+FP}$$$$Recall=\frac{TP}{TP+FN}$$$$F1\text{-}\mathrm{score}=\frac{2\times TP}{2\times TP+FP+FN}$$$$Accuracy=\frac{TP+TN}{TP+TN+FP+FN}$$

### 2.3. Support Vector Machine (SVM)

The data processing and model training pipeline was structured to evaluate Support Vector Machines (SVM) on a set of molecular descriptor datasets. The datasets were split into training and test sets, with test sets ranging from 20%, 25%, or 30% of the corresponding dataset. Stratified splitting was applied to preserve the class distribution. The data was standardized to ensure that all features were on a comparable scale. SVM models were trained using various kernel functions, including radial basis function (RBF), linear, polynomial, and sigmoid kernels. The models were evaluated using non-cross-validation (CV) and *k*-fold CV (with *k* = 5 or 10). The performance of each model was evaluated using accuracy as the metric, calculated as the proportion of correct predictions. Hyperparameters were performed by defining a list of potential values for the number of splits for *k*-fold CV, random states for CV and test split, test size splits, and kernel functions (kernel). Thus, we evaluated all combinations of these hyperparameters. Only the results with accuracy more or equal than 0.60 for the training set (acc_non−CV_ and acc_k−fold CV_) and accuracy more or equal than 0.50 for the test set (acc_pred_) were retained for ROC AUC analysis. The scripts are available in https://github.com/anacletosouza/Mpro_AI-ML_pipeline/ and Appendix A.

### 2.4. Logistic Regression (LR)

To construct and evaluate the Logistic Regression models applied to binary classification tasks derived from molecular descriptor datasets, we employed the same approach used for building the SVM models. However, the Logistic Regression models were trained using five distinct solvers: lbfgs, liblinear, newton-cg, sag, and saga, initially without cross-validation. The scripts are available in https://github.com/anacletosouza/Mpro_AI-ML_pipeline/ and Appendix A. The logistic function used in this study was defined by the following:
$$p(1\;|\;X)=\frac{1}{1+exp[-(\beta_{0}+\beta_{1}x_{1}+\beta_{2}x_{2}+\ldots +\beta_{n}x_{n}\;)]}$$
where the following is established:$$X{=(x}_{1},\ldots ,x_{n})$$

This denotes the vector of molecular descriptors used during the training of the Logistic Regression model.

### 2.5. Molecular Docking and Molecular Dynamics Simulations

The inhibitors, represented by the SMILES, were converted into a 3D structure using Chimera [28] tools, which was then used for molecular docking. The 3D structure of monomeric Mpro was built with AlphaFold3 [29]. We prepared each Mpro/inhibitor complex using the UCSF Chimera tool for molecular docking simulations, according to our previous studies [17]. Histidines 64, 163 were protonated in the ϵ2 tautomer; histidines were protonated in residues 41, 80, 164 as δ1 tautomer; charged histidines were protonated in 172 and 246. To build a Mpro/inhibitor complex, we docked each inhibitor into the binding site. Subsequently, the inhibitor was submitted to a systematic conformational assembly into the binding site using the autodocking Vina software [30,31]. The grid was centered in the Cys145 (SG, chain A, with coordinates (11.4, 12.1, −2.2) and size (40, 30, 30)). In the molecular docking parameters, we used exhaustiveness and number of modes of 8 and 1, respectively. We differed in the molecule poses, considering differences of 3 kcal/mol. The poses were evaluated by scoring representing the binding free energy. We performed nine consecutive minimizations of the Mpro/inhibitor complex, followed by MD runs performed in the Groningen Machine for Chemical Simulation Software (GROMACS, version 2022) [32,33], using optimized potentials for liquid simulations for all-atom (OPLS-AA) force fields. Inhibitor topology was built in the LigParGen web-based service [34,35,36], and the 1.14*CM1A charges were calculated considering neutral charge and 1 interaction. All systems were then explicitly solvated with TIP3P water models in a triclinic box and neutralized, keeping a 150 mM NaCl concentration, and minimized until reaching a maximum force of 10.0 kJ/mol or a maximum number of steps of 5000. The Mpro/inhibitor was equilibrated consecutively in isothermal-isochoric (NVT) and isothermal-isobaric (1 bar; NpT) ensembles, both at a temperature of 310 K for 2 ns (number of steps and intervals of 1,000,000 and 2 fs, respectively). All simulations were then performed in a periodic triclinic box, considering the minimum distance of 1.2 nm between any protein atom and triclinic box walls. Molecular dynamics runs were performed for 150 ns (number of steps and intervals of 75,000,000 and 2 fs, respectively) in NpT ensembles. We used the leapfrog algorithm to integrate Newton equations. We selected LINCS (LINear Constraint Solver) as a solver that satisfies holonomic constraints, whose number of iterations and order were 1 and 4, respectively. We used neighbor searching grid cells (cutoff scheme Verlet, frequency to update the neighbor list of 20 steps, and cutoff distance for the short-range neighbor list of 12 Å). In the van der Waals parameters, we smoothly switched the forces to zero between 10 and 12 Å. In electrostatic Coulomb, we used fast, smooth Particle-Mesh Ewald (SPME) electrostatics for long-range electrostatics. In addition, we defined the distance for the Coulomb cutoff of 12 Å, interpolation order for PME for a value of 4 (cubic interpolation), and grid spacing for Fast Fourier Transform (FFT) of 1.6 Å. In temperature coupling, we used velocity rescaling with a stochastic term (V-rescale; modified Berendsen thermostat), and after obtaining two coupling groups (Mpro/inhibitor and water/ions), we considered them for the time constant of 0.1 ps and 310 K as the reference temperature. In pressure coupling (NpT ensembles), we used a Parrinello–Rahman barostat (type isotropic, time constant of 1 ps, reference pressure of 1 bar, and isothermal compressibility of 4.5 × 10^−5^ bar^−1^). In the molecular dynamics calculations, we used periodic boundary conditions in xyz coordinates (3D space). Next, we calculated root-mean-square deviation (RMSD) and root-mean-square fluctuation (RMSF) values using GROMACS modules. To analyze structural dynamics of the Mpro/inhibitor complex, we extracted the Cα atomic coordinates of each residue from molecular dynamics trajectories and computed their Euclidean distances from the origin for each frame. The data were mean-centered, followed by Dynamical Cross-Correlation Matrix (DCCM) computation using Pearson correlation coefficients between atomic displacements. Correlated and anti-correlated motions were visualized with heatmaps and 3D structural networks in PyMOL (v. 2.5.0). From DCCM, we extracted eigenvalues and eigenvectors to identify normal modes of motion. Free Energy Landscapes (FELs) were derived from PCA scores using the relation ∆G = − RT ln(P + ξ), with ξ = 10−10 to prevent logarithmic singularities. Energies were binned (bin size = 50), normalized, and mapped back onto trajectory frames for visualization. To identify interactions within the subsites of Mpro, four main subsites were defined based on our previous study: S1 (containing PHE140, CYS145, HIS163), S1′ (THR25, THR26, HIS41, LEU27, ASN142, GLY143), S2 (MET49, TYR54, PRO52, HIS164, MET165, ASP187, ARG188, GLN189), and S3/S4 (MET165, GLU166, LEU167, PRO168, PHE185, THR190, ALA191) [5]. The analysis considered three distant atoms of the ligand: 3AN [C0F (quinoline), C05 (benzothiazole), O00 (thioamide)], 37 [C0F (pyrazole), C0O (chlorophenyl), C0T (benzotriazole)], 21 [C0R (benzotriazole), S07 (thiophene), N0H (imidazole)], and 40 [C0R (benzotriazole), N0K (imidazole), C0A (chlorophenyl)]. The coordinates of these atoms define the center of a sphere with a radius equal to 4 Å. All atoms within this sphere were considered potential interaction residues, and corresponding interacting subsites were assigned. Based on that, the contact frequency per residue and per subsite was calculated. Scripts used in these methodologies are available in Appendix A and https://github.com/anacletosouza/Mpro_AI-ML_pipeline/. Simulations and analyses were run on an HPC cluster comprising four nodes equipped with high-memory CPUs and Nvidia GPUs (L40S and A30) (Institute of Biomedical Sciences, University of São Paulo, São Paulo city, São Paulo state, Brazil), ensuring efficient large-scale data processing.

## 3. Results

### 3.1. Support Vector Machine (SVM) Models

In this study, we adopted a systematic approach to build and evaluate SVM-based classification models using different datasets and molecular descriptors in the feature selection process. The datasets varied in the IC_50_ measurement method (FRET, fluorescence, or a combination of FRET/fluorescence/SPR), the type of molecular descriptors (2D or 3D), and the filtering thresholds applied based on the pairwise Pearson correlation between the numerical descriptors. We performed multiple hyperparameter combinations, adjusting the number of cross-validation splits (k-fold), the training and test set splits, the random seed values, and the kernel types (linear, radial basis function, polynomial, and sigmoid). For each combination, the SVM models were trained and evaluated using the metrics of accuracy, recall, precision, F1-score, and area under the curve of the ROC curve (ROC AUC). Then, the performance of these models was evaluated using the same metrics in the test set. In total, we built and evaluated 1584 models, of which 1204 presented accuracy without cross-validation (CV) (acc_non−CV_), accuracy with cross-validation (acc_k−fold CV_), and prediction accuracy (acc_pred_) values greater than or equal to 0.6, 0.6, and 0.5, respectively. This strategy allowed us to explore the hyperparameter space and select the best models for solving classification problems.

Next, we selected the three best models constructed from each dataset, considering only those with acc_non−CV_ and acc_pred_ greater than 0.7, in addition to prioritizing those that presented the largest AUC ROC in both the training and test sets (Table 1). The best model (Model 9) was obtained using the set of molecules whose IC_50_ measurement method was by fluorescence and 36 molecular descriptors (condition 43, Appendix A). Molecular descriptors used in the construction of model 9 are shown in Appendix A, and parameters and hyperparameters are summarized in the Appendix A. This SVM model was built with radial basis function (RBF), 5- and 10-fold cross-validation, a random seed for k-fold CV (kf) of 37, a test size of 0.2 (319 compounds), and a training size of 1595 compounds, using a random seed (split) of 21. This SVM classifier utilized 769 support vectors—357 corresponding to active and 412 to inactive compounds—and a gamma value (for RBF) of ~0.03. In the test set, the model achieved an accuracy of 0.79 and a precision of 0.75, indicating reliable overall classification performance. This pattern is corroborated by the confusion matrix, which showed 79 true positives and 26 false negatives; in contrast, the classification of inactive compounds was more robust, with 174 true negatives and 40 false positives (Figure 2). This disparity may be related to class imbalance, as the training set contained 475 active and 801 inactive compounds, and the test set included 119 active and 200 inactive compounds. Despite this limitation, the ROC-AUC value of 0.86 indicates strong discriminative ability, supporting the model’s overall suitability for identifying active and inactive Mpro inhibitors. In contrast, models based on IC_50_ measured by FRET presented high recall (close to 1) but lower precision (between 0.72 and 0.73) and SVM models built from combined dataset (FRET, fluorescence, and SPR) presented lower ROC-AUC values (between 0.20 and 0.28), showing limited capacity to separate active and inactive inhibitors. This result suggests that, even in high data dimensionality—generally favorable for SVMs—the mixture of measurement methods impairs model performance.

### 3.2. Logistic Regression (LR) Models

*Model description.* The Logistic Regression (LR) models were built and evaluated with a focus on interpretability and generalization, both assessed through performance metrics on training and test sets. Table 2 presents the performance of LR models. Model 1 (FRET, condition 14, 47 molecular descriptors—Appendix A) achieved a recall of ~0.97, ROC AUC was ~0.71, and acc_non−CV_ of ~0.75 for the test set was considered satisfactory in our analysis. Conversely, models 10–13 (condition 45 in Appendix A, based on fluorescence data, using 85 molecular descriptors)—trained and tested with a random split of 75% (1196 inhibitors) and 25% (399 inhibitors), respectively—exhibited superior generalization performance. These models reached AUCs of ~0.85 and ~0.83 on the training and test sets, respectively. The average accuracies under 5-fold and 10-fold CV (acc_5−fold CV_ = 0.78; acc_10−fold CV_ = 0.76) were consistent across training and test sets, suggesting that the models learned stable and generalizable patterns. Although the recall (0.60) was lower than that of the models generated under condition 14, the overall predictive performance was more robust. The models built under condition 31 (integrating FRET, fluorescence, and SPR data, with 81 descriptors) showed intermediate performance compared to the previously described models, with AUCs ranging from 0.76 to 0.77. The accuracies under 5-fold and 10-fold CV ranged from 0.73 to 0.74, and test set accuracies ranged from 0.71 to 0.72. These results indicate that, although all these LR models exhibit acceptable generalization capabilities, models 10–13 stand out in this regard. Given that models 10–13 differ only in the method used for estimating the Logistic Regression coefficients, model 10 was selected for subsequent analyses due to its consistent and superior generalization performance. Specifically, model 10 (Logistic Regression with L2 regularization, newton-cg solver) was trained using 85 molecular descriptors covering diverse physicochemical and topological features (Table 2 and Appendix A).

The parameters and hyperparameters of model 10 are summarized in the Appendix A. The model employed an adjusted intercept (∼0.85), L2 regularization (C = 1.0), and a convergence tolerance of 0.0001, reaching the stopping criterion in eight iterations (maximum: 1000). Performance metrics indicated robust generalization, with training accuracy (~0.78) closely matching test accuracy (~0.76). Precision exceeded recall in both sets (training: ~0.73; test: ~0.71 vs. training recall: ~0.66; test recall: ~0.60), while F1-scores were ~0.69 (training) and ~0.66 (test). The strong ROC-AUC scores (~0.85 training, ~0.83 test; Figure 3) further confirmed discriminatory power between active and inactive classes. As shown in Figure 3, the confusion matrices presented consistency in the training/test ratio with 294/90 true positives, 151/59 false negatives, 111/36 false positives, and 640/214 true negatives (Figure 3). Since the coefficients (provided in the Appendix A) quantify each descriptor’s contribution to predicting the active class, and the consistent performance between training and test distributions demonstrates minimal overfitting, this Logistic Regression model emerges as the optimal choice for reliable coefficient interpretation. The model’s robust performance across all evaluation metrics further supports its suitability for the mechanistic interpretation of descriptor contributions in binding affinity of the Mpro inhibitors. In contrast to the SVM model, the Logistic Regression (LR) model allows interpretability, where weights associated with Logistic Regression indicate positive contributions (positive coefficient values) and negative contributions (negative coefficient values) toward a compound being classified as active. In this regard, we considered the coefficients obtained from the standardized data. Thus, to investigate how pharmacokinetic properties relate to biological activity, we compared the contribution of each coefficient representing a molecular descriptor with the probability of the inhibitor being active. The selected descriptors in this discussion are involved with Lipinski’s Rule [37], TPSA [38], and QED [39].

*Drug-likeness profile.* Other molecular descriptors used in the construction of Model 10 are shown in Appendix A. Lipinski’s Rule is widely used in the early screening of drug-like compounds to predict oral bioavailability compounds with good intestinal absorption tend to violate no more than one of the following empirical criteria: molecular mass ≤ 500 Da, octanol/water partition coefficient (log *P*) ≤ 5, ≤5 hydrogen bond donors (OH and NH groups), and ≤ 10 hydrogen bond acceptors (oxygen and nitrogen atoms) [37]. These parameters are associated with physicochemical properties favorable to membrane permeability and aqueous solubility. Beyond Lipinski’s criteria, topological polar surface area (TPSA) is another key parameter for predicting oral absorption. TPSA represents the sum of polar surface areas from chemical functionalities—such as oxygen and nitrogen atoms—and correlates with a compound’s ability to cross cell membranes. Generally, compounds with TPSA < 140 Å^2^ are more likely to be well-absorbed orally [38], while values below 90 Å^2^ are considered favorable for central nervous system penetration [40]. The quantitative estimate of drug-likeness (QED) is particularly important during the hit-to-lead and lead optimization phases because it captures the balance among key features such as molecular weight, lipophilicity, polarity, solubility, and synthetic accessibility—interpretable score ranging from 0 (poor drug-likeness) to 1 (optimal drug-likeness) [39]. In this context, the relatively low QED value highlights the importance of further rational modifications to enhance pharmacokinetic compatibility while retaining biological activity. In this regard, combining Lipinski’s Rule with TPSA and QED thresholds provides a strategic approach to selecting compounds with potential for good oral bioavailability.

*Available inhibitors have a hydrophilic profile with a penalty in PK properties*. Based on that, we observe that the model coefficients indicate that certain pharmacokinetic properties, often associated with good absorption and distribution, may reduce the probability of a compound being active against Mpro. For example, the negative coefficient for MolLog *P* (~−0.6) suggests that hydrophilic compounds tend to be more active. This contrasts with Lipinski’s Rule, which favors log *P* ≤ 5 for good permeability. This implies that reducing lipophilicity increases the probability of activity, possibly due to interactions with hydrophilic residues located in the Mpro S1′ and S1 subsites, which are regions with more polar physicochemical characteristics. Additionally, TPSA exhibited a negative coefficient approaching zero (−0.000156), indicating that this parameter has minimal influence on activity prediction. This suggests that Mpro may tolerate molecules with higher polarity without compromising activity, differing from more hydrophobic targets.

Another notable aspect is the positive coefficient for ExactMolWt (~0.37), indicating that heavier molecules tend to be more active. This may be related to the need for more extensive interactions with Mpro, aiming to enhance inhibitors that maximize the number of contacts in a large and complex active site. Conversely, the negative coefficient for LabuteASA (~−0.77) suggests that molecules with a smaller solvent-accessible surface area are favored. Considering the hydrophobic, aromatic, and hydrogen bonding natures of the subsites S2 and S3/S4, these features may indicate that these subsites could be further exploited to improve both pharmacodynamic and pharmacokinetic properties of these inhibitors. Furthermore, descriptors such as BertzCT (~0.44) and Kappa3 (~0.53) have positive coefficients, indicating that greater molecular complexity and higher structural rigidity may favor activity. SlogP_VSA2 (~−0.55) and VSA_EState2 (~−0.51) reinforce that properties related to highly lipophilic regions or certain electronic states may decrease binding affinity to Mpro.

The QED coefficient for the current LR model was ~0.37, indicating that QED may be explored for structural optimization of inhibitors. Therefore, these results suggest that while properties such as log *P* and TPSA within conventional ranges are typically sought to optimize pharmacokinetics, a more flexible approach may be necessary for Mpro. Excessive increase in lipophilicity to satisfy drug-likeness rules may compromise target interaction, whereas higher TPSA does not appear to be a limiting factor. Thus, the discovery of effective Mpro inhibitors may require a different balance between pharmacokinetics and pharmacodynamics, prioritizing structural features that favor target binding even if this entails partial deviations from traditional permeability rules. This underscores the importance of target-specific predictive models, avoiding generalizations that could lead to false negatives in virtual screening.

*Comparison with FDA-approved drugs.* We applied the RDkit functions on the protonated molecules to calculate the 36 descriptors used in the construction of the SVM model (Appendix A) and the 85 descriptors used in the LR model (Appendix A) for Boceprevir, Ensitrelvir, Lufotrelvir, Nirmatrelvir, Ritonavir, and Simnotrelvir (active Mpro inhibitors in preclinical or clinical phases). From these descriptors, we constructed a 95% interval of confidence (IC95%) for the descriptors used in these models to establish the drug-likeness profile for molecules in advanced development stages. We included Ritonavir due to its inhibitory activity against Mpro (IC_50_ = 13.7 µM and EC_50_ = 19.9 µM) and its favorable pharmacokinetic properties (it is a component of Paxlovid) [5,41,42]. Appendix A present the mean, standard deviation, and confidence interval (α = 0.05) for the 35 and 85 descriptors of these inhibitors. For these Mpro inhibitors, the means of ExactMolWt, QED, TPSA, and MolLog *P* were respectively 561.56 Da [IC95% = (477.5 − 645.6)], 0.33 [IC95% = (0.18 − 0.48)], 146.4 Å^2^ [IC95% = (0.18–0.48) Å^2^], and 2.03 [IC95% = (−0.2 − 4.2)]. The means, standard deviations, and confidence intervals for other descriptors are provided in Appendix A for these inhibitors. Nirmatrelvir and Simnotrelvir showed the highest QED values of 0.50 and 0.46, respectively, while Boceprevir, Ritonavir, and Lufotrelvir exhibited values of 0.36, 0.11, and 0.24, respectively. All compounds displayed comparable molecular weights (MolWt) with a mean of 562 Da and a standard deviation of 80 Da [IC95% = 477.9 − 646.2 Da]. The minimum and maximum partial charges had mean values and standard deviations of −0.4 ± 0.2 and 0.3 ± 0.1, respectively. The confidence intervals, as well as mean values and standard deviations of the molecular descriptors, are available in Appendix A. These results are fundamental for hit-to-lead optimization studies and for high-throughput virtual screening (HTVS) to identify novel hits.

### 3.3. Molecular Docking and Molecular Dynamics Simulations

We performed molecular docking followed by molecular dynamics (MD) simulations in order to investigate the interactions of Mpro inhibitors at the active site, aiming to elucidate how their pharmacodynamic and structural features may help to improve pharmacokinetic behavior. To that end, 150 ns MD simulations were carried out for a subset of randomly selected inhibitors from a dataset with IC_50_ values determined by fluorescence-based assays (Figure 4 and Figure 5).

*Mpro S2 and S3/S4 may be a potential pharmacophoric center*. Overall, the high numbers of correlations and anti-correlations are directly linked to inhibitor activity in Mpro. Potent inhibitors are more stable in the active site, maintained by hydrogen bonding and van der Waals interactions. In general, these potent inhibitors stabilize domains I and II, reducing their flexibility and inducing allosteric effects in other regions of the enzyme, as shown in RMSD and RMSF plots (Appendix A). This is evidenced by strong correlations and anti-correlations between residue pairs across different domains, as shown in Figure 4. Examples of potent inhibitors following this pattern include inhibitors 37 (2163 correlations and 1029 anti-correlations; IC_50_ of 0.33 μM). These molecules are characterized by performing aromatic, hydrogen bonding, and hydrophobic interactions in the subsites S2 and S3/S4. These findings suggest that subsites S2 and S3/S4 may be a potential pharmacophoric center to be explored in hit-to-lead optimization.

We also investigated the free-energy landscape (FEL) (Figure 5) and potential interactions (Figure 6) at the different sites of Mpro during binding with inhibitors 3AN, 37, 21, and 40. In relation to FEL results, the number of blue points (favorable free-energy values) (compared with red points that represent unfavorable free-energy values) in the PCA plot for each inhibitor is in agreement with the corresponding experimental IC_50_ value of them. In relation to potential interactions, our analysis focused on key residues in subsites S1 (PHE140, CYS145, HIS163), S1’ (THR25, THR26, LEU27, HIS41, ASN142, GLY143), S2 (MET49, PRO52, TYR54, HIS164, MET165, ASP187, ARG188, GLN189), and S3/S4 (MET165, GLU166, LEU167, PRO168, PHE185, THR190, ALA191), as defined in our previous study [5]. Contact frequencies were quantified for each residue and subsite, and amino acid residues interacting with these inhibitors were identified (Figure 6), enabling a systematic assessment of interactions between inhibitor regions, Mpro, and the involved amino acids. Our analyses revealed interaction preferences of these inhibitors in subsites S2 and S3/S4. For instance, the quinoline group of 3AN interacted predominantly with subsite S3/S4 (1355 occurrences) compared to S2 (643 occurrences), while the benzothiazole group preferentially interacted with S2 (1799 occurrences) rather than S3/S4 (618 occurrences). At the residue level (Appendix A), quinoline primarily interacted with MET49, LEU50, GLN189, and THR190, whereas benzothiazole had more contacts with SER46, MET49, GLN189, and THR190. These results suggest permanent dipole–induced dipole interactions and potential hydrogen bonding mediated by the heteroatoms of the aromatic system of this chemical group. In addition, the thioamide group interacted predominantly with MET39, SER46, GLN189, and THR190. This pattern highlights the central role of the C(=S)NH moiety in establishing hydrogen bonding and van der Waals interactions. Thus, interactions identified in the 3AN/Mpro complex highlighted specific subsites S2 and S3/S4 as critical interaction regions. On the other hand, inhibitors 21, 37, and 40 possess a benzotriazole group responsible for interacting with subsites S1’, S1, S2, and S3/S4 (Figure 6B–D). However, the pyrazole group (inhibitor 37) specifically interacted with subsite S1’ (2259 occurrences) (primarily with ASN119, THR26, TYR118, GLN19, GLY143), being more solvent-exposed, whereas the imidazole group of inhibitor 21 preferentially contacted subsites S2 (1358 occurrences), S3/S4 (543 occurrences), and S1’ (129 occurrences) (main contacts with SER46, LEU50, GLN189, GLU47, PRO168, MET49). The imidazole group of inhibitor 40 was also more solvent-exposed (1593 occurrences in S1’) and interacted less with S3/S4 (512 occurrences) (preferentially with THR25, ASN142, GLY143, THR24).

Overall, this may explain why compounds 37 and 40 exhibit lower potency (IC_50_ of 0.33 and 0.17 μM, respectively), whereas 3AN and compound 21 show higher binding affinity with Mpro (0.09 and 0.06 μM, respectively). All these are in agreement with our FEL results (Figure 5). Considering that the PK/PD trade-offs are fundamental for hit-to-lead optimization and that, according to our ML data, the most potent inhibitors are those with more hydrophilic characteristics (contrary to PK properties) and vice versa. Additionally, assuming that MD simulation data were substantially important for identifying the S2 and S3/S4 subsites, as well as the mean and confidence interval data of advanced-stage inhibitors discussed here, we suggest that inhibitors with the properties listed in Appendix A, combined with the confidence interval and MD simulation data, may provide a basis for selecting new hits and lead optimization. This is based on the fact that the most potent inhibitors interact with the S2 and S3/S4 subsites. Thus, these results provide a rational direction for the search for new SARS-CoV-2 Mpro inhibitors.

## 4. Discussion

The rising costs and extended timelines associated with conventional drug development have positioned computational approaches as an increasingly attractive strategy in pharmaceutical research [43,44]. By capitalizing on compounds with well-characterized safety profiles, this approach significantly reduces clinical risks while providing expedited and cost-efficient routes to therapeutic discovery [45,46,47]. Molecular docking and dynamics simulations have enabled systematic evaluation of diverse chemical scaffolds against key viral targets, facilitating rapid identification of high-affinity binders [48]. The integration of pharmacophore modeling with molecular dynamics has further refined this process, yielding several FDA-approved drug candidates with predicted activity against SARS-CoV-2 main protease (Mpro) [49,50]. Despite these advances, a critical bottleneck persists, while numerous compounds show promising in silico activity against Mpro, few progress to clinical evaluation [5]. This translational gap may often be related to pharmacokinetic (PK) limitations, as exemplified by Boceprevir [5]. Although this HCV protease inhibitor demonstrates anti-SARS-CoV-2 activity (IC_50_ = 4.13 µM, EC_50_ = 1.31 µM), its clinical potential remains constrained by CYP3A4 interactions. The successful development of Paxlovid (Nirmatrelvir/Ritonavir) illustrates how strategic PK optimization may overcome these barriers [5].

In this study, we investigated the relationship between pharmacokinetic (PK) and pharmacodynamic (PD) properties using integrated machine learning (ML) and molecular dynamics (MD) approaches. We developed binary classification models employing multiple ML algorithms—Support Vector Machines (SVM) and Logistic Regression (LR). These models were trained on IC_50_ data from FRET assays, fluorescence assays, and combined datasets (FRET, fluorescence, and SPR assays). These computational approaches use large datasets and predictive algorithms to rapidly evaluate chemical compounds, accelerating the discovery of promising candidates with inhibitory activity against SARS-CoV-2 [51,52,53]. Other QSAR models have been developed to identify SARS-CoV-2 Mpro inhibitors. For example, convolutional neural networks (CNNs) were used to predict pIC_50_ values from a dataset of 467 molecules, having a training accuracy of 95.7% and a test accuracy of 90.43% [54] while random forests (RF) classified compounds as active or inactive using 940 samples, with a training accuracy of 89.8% and test accuracy of 64.2% [55]. On the other hand, partial least squares (PLS) regression was applied to predict pIC_50_ values from a smaller dataset of 32 samples, resulting in a training accuracy of 90.1% and a test accuracy of 83.1% [56], support vector regression (SVR) predicted pIC_50_ values for a dataset of 71 samples, achieving 96.0% training accuracy and 89.3% test accuracy [57], and artificial neural networks (ANNs) were employed to predict p*K*_i_ values from 28 samples, reaching a training accuracy of 80.0% and a test accuracy of 91.0% [17]. This comparison highlights the variability in performance across different models and datasets (containing different approaches and sizes), emphasizing the need for careful selection of methods based on specific prediction tasks.

The main limitation across the compared machine learning methods is the variability in performance, driven by sample size, model complexity, and data representativeness. While complex models such as deep neural networks achieve high training accuracy, their generalization may be inconsistent, particularly with small datasets. Methods such as random forests also show limited generalizability, highlighting the need for model selection in order to improve the prediction of novel SARS-CoV-2 Mpro inhibitors [58,59]. Our Support Vector Machine (SVM) and Logistic Regression (LR) models, trained on chemically diverse datasets, demonstrated robust performance in distinguishing active from inactive compounds, achieving strong predictive accuracy validated by consistent ROC AUC values across both training and test sets. Recent advances have demonstrated the powerful synergy of machine learning (ML), molecular docking, and molecular dynamics (MD) in studying SARS-CoV-2 drug candidates, enabling simultaneous evaluation of structure-activity relationships and pharmacokinetic properties [17,60,61,62]. These approaches have successfully identified diverse chemical scaffolds with high predicted affinity for viral targets, as exemplified by evaluation of ten Mpro inhibitors exhibiting favorable drug-like properties (molecular weights 383.15–534.18 Da, logP~2.9–5.3, TPSA 55.11–113.84 Å^2^, and three-to-seven rotatable bonds) [60]. The integration of computational techniques—particularly pharmacophore modeling with MD simulations has accelerated drug repurposing efforts by efficiently prioritizing stable, high-affinity binders among FDA-approved compounds [49,50].

Our results are supported by Nirmatrelvir (PF-07321332) (Figure 1). The binding to the S3 and S4 subsites of Mpro validates the relevance of these regions for antiviral activity—a finding consistent with our molecular dynamics simulations, which revealed dynamic correlations between the S2 and S3/S4 [63]. Furthermore, Nirmatrelvir exhibits a balanced profile of hydrophilicity (LogP~2.9) and oral bioavailability, reinforcing our Logistic Regression-based hypothesis that more hydrophilic compounds tend to be active, although they require pharmacokinetic optimization. Recent structural studies by Blankenship and co-authors provide direct experimental support for our mechanistic insights, showing that inhibitor analogs to Nirmatrelvir, such as MPI105 (IC_50_ = 0.039 μM), achieve high potency targeting S3, and MPI108 (IC_50_ = 0.025 μM) binds at S3/S4 [64]. These findings further corroborate our conclusion that effective inhibitor design depends on integrating interactions across S2 and S3/S4.

Our study advances this field by establishing, for the first time, a structure-based framework for PK optimization of Mpro inhibitors through integrated ML/MD analysis. Key insights reveal that while hydrophilic features enhance target binding, they often impair PK properties, whereas hydrogen bonding, hydrophobic, and π–π interactions in subsites S2 and S3/S4 stabilize the inhibitor contacts into active sites of Mpro. This finding highlights the subsites S2 and S3/S4 as a strategic target for achieving optimal PD/PK balance. Therefore, our computational pipeline provided predictive models for classification in inactive and active compounds against Mpro and used structure-based design principles for Mpro inhibitors with applications in the preclinical candidate selection. These advances address a critical gap in current computational drug discovery by enabling hit-to-lead optimization. This integrative approach not only enhances Mpro inhibitors but also provides a transferable framework for addressing other molecular targets through structure-guided computational design.

## 5. Conclusions

Herein, we used a computational approach that integrates ML algorithms, molecular docking, and MD simulations to explore the often conflicting relationship between PD potency and PK in the context of designing inhibitors targeting the SARS-CoV-2 Mpro. The ML models employed in this framework, specifically Support Vector Machine (SVM) and Logistic Regression (LR), showed strong predictive capabilities in distinguishing active Mpro inhibitors from inactive compounds. The SVM model achieved a test accuracy of 0.79 with an area under the ROC curve (AUC) of 0.86, while the LR model closely followed with an accuracy of 0.76 and an AUC of 0.83. These results underscore the potential of data-driven methods in early-stage drug discovery, particularly when predicting activity across diverse chemical scaffolds.

Through detailed analysis, it was observed that certain molecular features—especially those associated with hydrophilicity, such as lower MolLog *P* values—tended to correlate with higher binding affinity to Mpro. Conversely, properties typically favorable for PK profiles, such as increased lipophilicity and reduced molecular weight, were frequently found to compromise binding potency. This observation illustrates a fundamental trade-off between optimizing a compound for effective target engagement and ensuring it possesses suitable drug-like characteristics for clinical development. Molecular dynamics simulations provided further mechanistic insights by revealing the importance of specific interaction regions within the protease structure. In particular, the Mpro S2 and S3/S4 subsites emerged as key for stabilizing inhibitor binding, mediated primarily by a combination of hydrogen bonding, hydrophobic contacts, and π–π stacking interactions. These analyses, including dynamical cross-correlation matrices and free-energy landscapes, indicated that potent inhibitors tend to induce conformational stabilization in the protease, particularly by reducing the flexibility of domains I and II, while potentially promoting long-range allosteric effects that may further influence enzymatic activity.

An exploration of molecular descriptors highlighted how features linked to chemical complexity—such as ExactMolWt, BertzCT, and Kappa3—correlated with increased biological activity. However, these same descriptors often conflict with attributes that favor desirable PK outcomes, such as oral bioavailability and metabolic stability. This intrinsic tension complicates the development process, as enhancing one aspect of a molecule’s profile frequently detracts from another. Nonetheless, a few advanced-stage inhibitors currently in clinical evaluation, such as Nirmatrelvir and Simnotrelvir, exemplify an emerging class of compounds that manage to strike a workable balance between PD and PK demands. Their quantitative estimates of drug-likeness (QED scores ranging from 0.33 to 0.50) and topological polar surface areas (around 146 Å^2^) serve as practical reference points for future optimization efforts.

Overall, this study presents an interpretable, predictive, and integrative computational strategy aimed at enhancing the PK/PD properties of antiviral drug candidates targeting SARS-CoV-2 Mpro. By focusing on structurally relevant subsites of the Mpro enzyme, particularly S2 and S3/S4, the framework not only pinpointed high-potential binding regions for drug optimization but also revealed critical insights that can drive the next generation of Mpro inhibitors. Additionally, the developed machine learning models demonstrated strong potential for large-scale virtual screening of novel chemical entities (NCEs), offering a robust platform for accelerating hit-to-lead discovery. Future efforts should prioritize the experimental validation of top computational hits, expand the chemical space through fragment-based design strategies that emphasize PK/PD synergy, and integrate resistance profiling to ensure sustained efficacy against emerging SARS-CoV-2 variants.

## Figures and Tables

**Figure 1 viruses-17-00935-f001:**
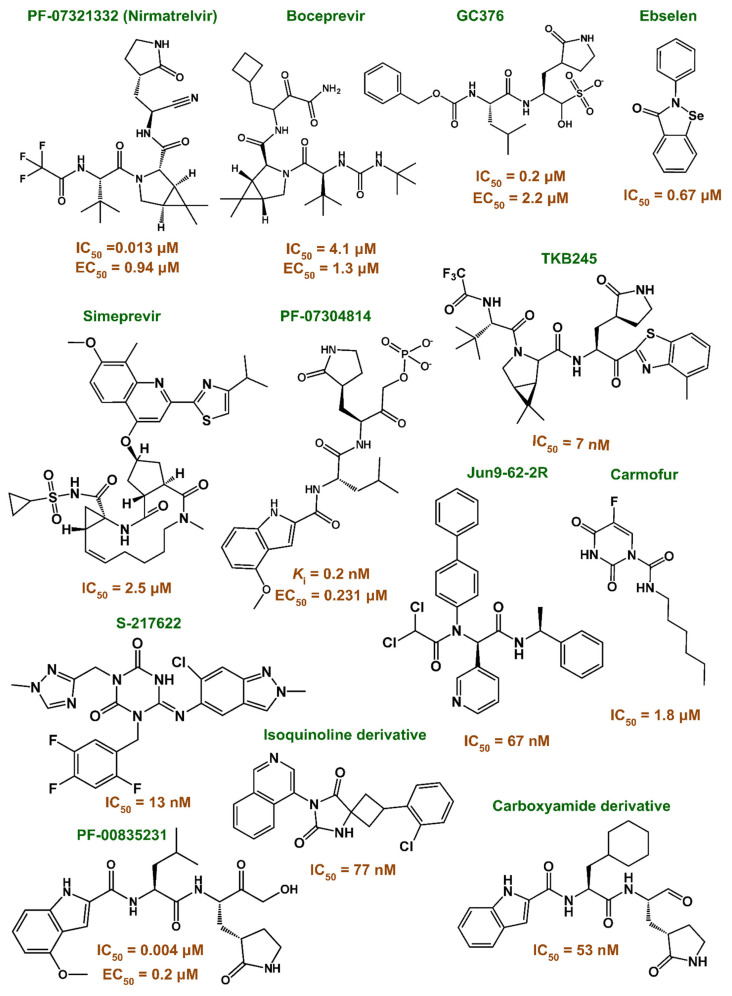
Chemical diversity and potency of Mpro inhibitors. The broad active site of Mpro supports the development of diverse inhibitors with high potency (IC_50_ between 0.3 and 2500 nM). This chemical diversity offers significant potential for anti-SARS-CoV-2 drug development.

**Figure 2 viruses-17-00935-f002:**
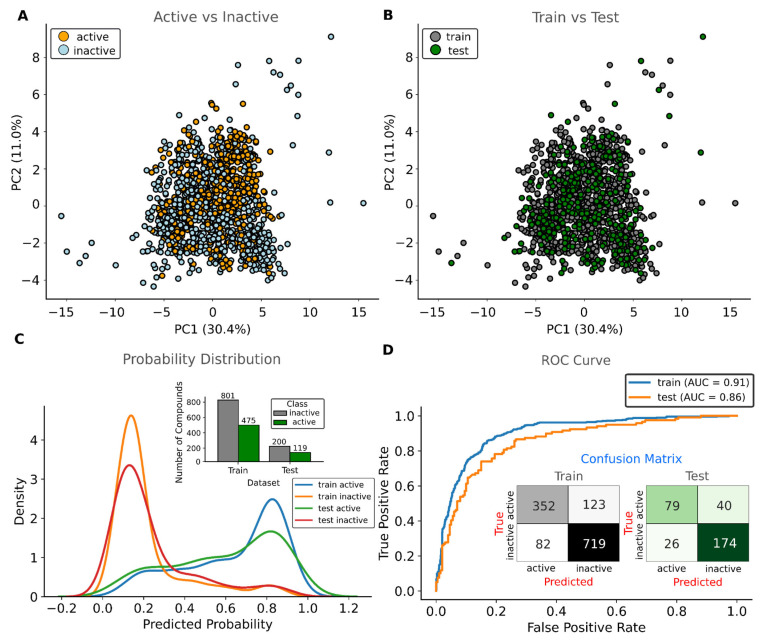
Chemical diversity and performance of the Support Vector Machine model (Model 9). (**A**) Distribution of active and inactive compounds in the PCA chemical space. (**B**) Random partitioning of compounds into training and test sets for SVM model construction, preserving overall data distribution. (**C**) Probability density of predicted class assignments from the SVM classifier. (**D**) Receiver operating characteristic (ROC) curve depicting SVM performance in distinguishing active from inactive compounds.

**Figure 3 viruses-17-00935-f003:**
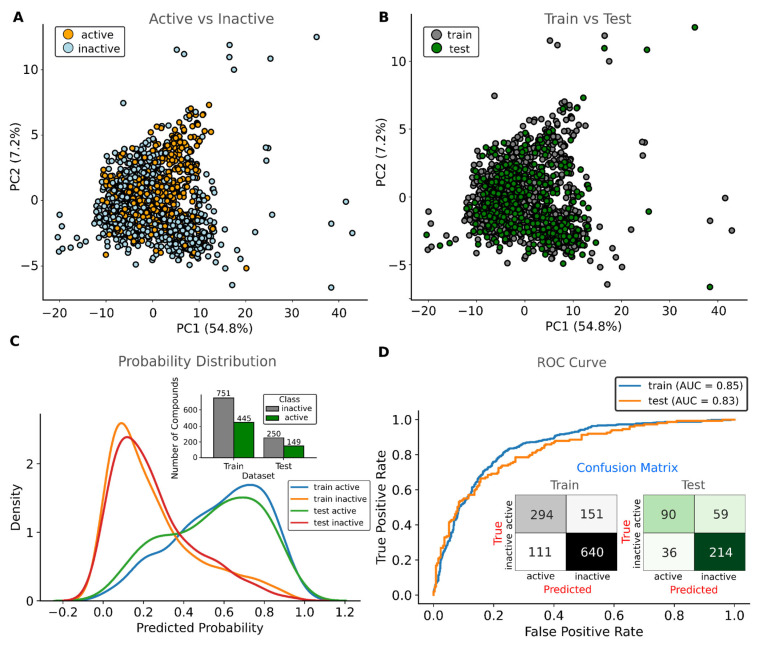
Chemical diversity and performance of the Logistic Regression (LR) model (Model 10). (**A**) Distribution of active and inactive compounds in the PCA chemical space. (**B**) Random partitioning of compounds into training and test sets for LR model construction, preserving overall data distribution. (**C**) Probability density of predicted class assignments from the LR model. (**D**) Receiver operating characteristic (ROC) curve depicting LR performance in distinguishing active from inactive compounds.

**Figure 4 viruses-17-00935-f004:**
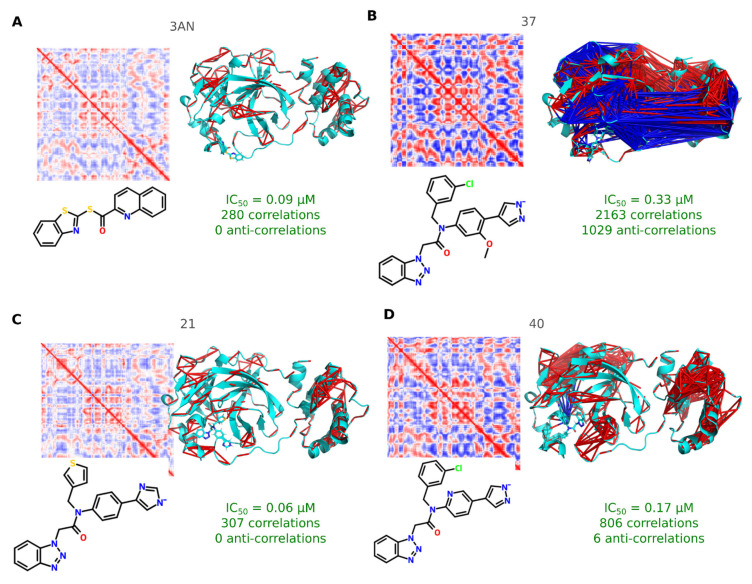
Structural analysis of the Mpro interacting with random inhibitors using dynamical cross-correlation matrix (DCCM). Dynamical cross-correlation matrix (DCCM) calculated using pairwise Pearson correlations of C_α_–C_α_ distances for each residue in the Mpro. The correlation matrix has dimensions 297 × 297, reflecting the correlation between residue pairs, where blue represents anti-correlated residue pairs while red represents correlated residue pairs. DCCM was filtered, keeping correlations more than 0.8 and anti-correlations less than −0.8. These correlations and anti-correlations were projected into the Mpro/inhibitor complex structure (after NpT equilibrium, *t* = 0 ns). The inhibitors considered in these analyses include: (**A**) 3AN; (**B**) 37; (**C**) 21; (**D**) 40.

**Figure 5 viruses-17-00935-f005:**
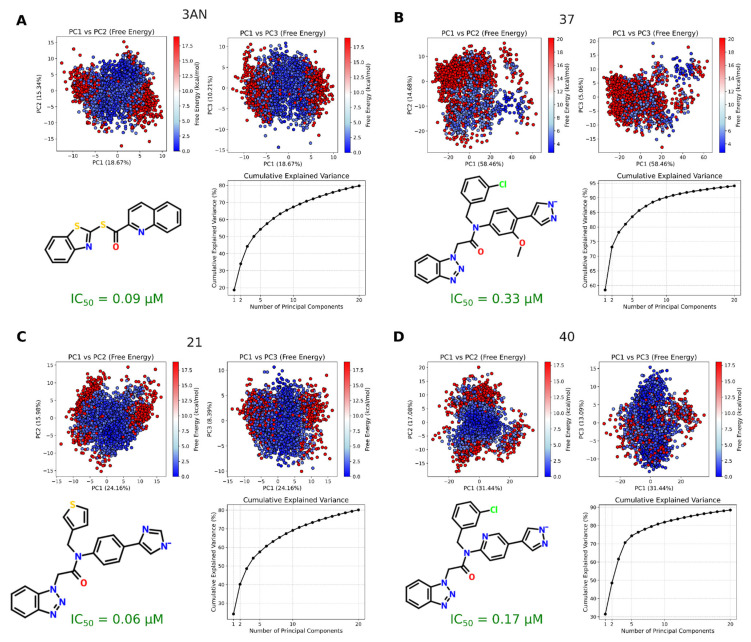
Principal component analysis (PCA) and free-energy landscape (FEL) of SARS-CoV-2 Mpro conformational dynamics. PCA considers the projection of C_α_ distances for each residue over time onto the PC1, PC2, and PC3 eigenvectors. The FEL heat map illustrates the evolution of energy changes associated with conformational transitions of SARS-CoV-2 Mpro along the PC1–PC2 and PC1–PC3 projected trajectories. The data are presented for the following inhibitors: (**A**) 3AN; (**B**) 37; (**C**) 21; (**D**) 40. For each case, PC1 explains 18.67, 58.46, 24.16, and 31.44% of the data variability, respectively; PC2 explains 15.34, 14.68, 15.98, and 17.08%, respectively; and PC3 explains 10.21, 5.06, 8.39, and 13.09%, respectively. Additionally, a cumulative variance plot was generated to depict the contribution of each principal component to the total variance. The FEL, obtained via PCA, represents the system’s free energy landscape in relation to its principal coordinates, highlighting stable states, conformational transitions, and energetic barriers. The FEL was calculated by generating a two-dimensional histogram of the PC1 and PC2 scores, with the corresponding probability density converted into free energy using the equation ∆G = − RT ln(P + ξ), where ∆G is the free energy, R is the gas constant, T is the absolute temperature (310 K), P is the probability density, and ξ = 10^−10^ is a regularization term to avoid singularities. Each point in the PC1–PC2 space corresponds to a free energy value based on its bin (bin size = 50) in the 2D histogram. Blue regions indicate low-energy (stable) conformational states, whereas red regions denote high-energy barriers separating these states. The pathways between these valleys suggest possible routes for conformational transitions.

**Figure 6 viruses-17-00935-f006:**
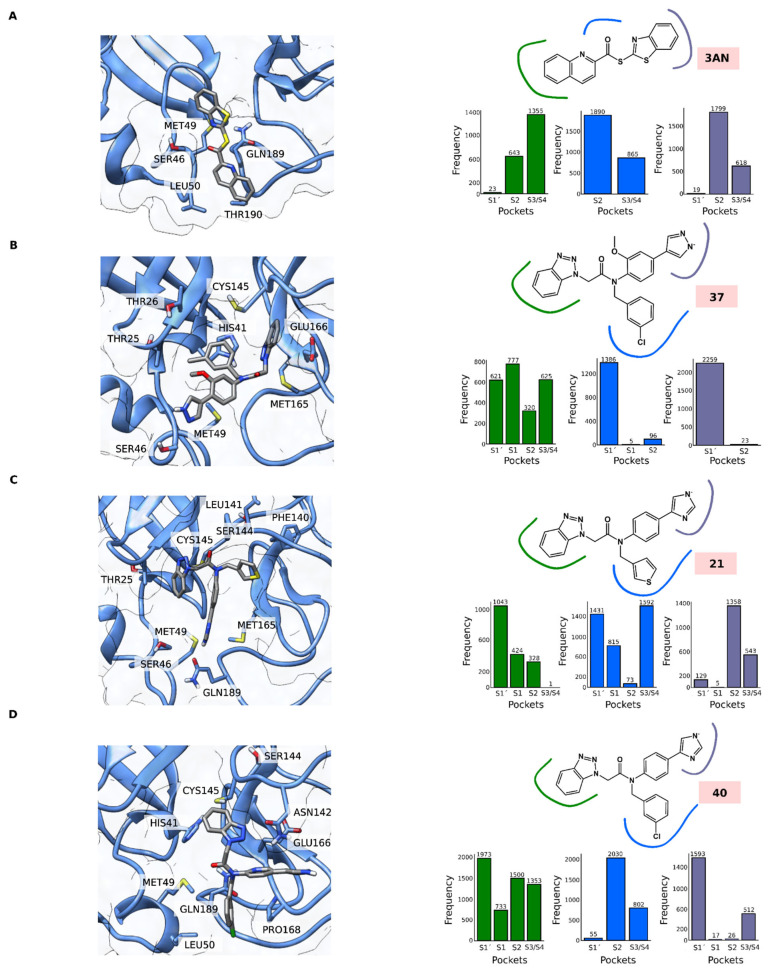
Potential interactions of SARS-CoV-2 Mpro with different inhibitors. To identify the interactions in the subsites of Mpro, four main subsites were defined in the protein based on our previous study [5]: S1 (containing PHE140, CYS145, HIS163), S1’ (THR25, THR26, HIS41, LEU27, ASN142, GLY143), S2 (MET49, TYR54, PRO52, HIS164, MET165, ASP187, ARG188, GLN189), and S3/S4 (MET165, GLU166, LEU167, PRO168, PHE185, THR190, ALA191). This analysis considered three distant atoms (and different chemical groups) that compose the inhibitor: 3AN [C0F (quinoline), C05 (benzothiazole), O00 (thioamide)], 37 [C0F (pyrazole), C0O (chlorophenyl), C0T (benzotriazole)], 21 [C0R (benzotriazole), S07 (thiophene), N0H (imidazole)], and 40 [C0R (benzotriazole), N0K (imidazole), C0A (chlorophenyl)]. The coordinates of these atoms define the center of a sphere with a radius of 4 Å. All atoms within the sphere were considered potential residues for interaction, and the subsite with which the inhibitor interacted was assigned. Next, the contact frequency per residue and per subsite was calculated. The data of the interactions are presented for the following inhibitors: (**A**) 3AN; (**B**) 37; (**C**) 21; (**D**) 40.

**Table 1 viruses-17-00935-t001:** Performance of Support Vector Machine models. The condition column specifies the descriptors computed and included in model construction, as detailed in Appendix A. Kernel functions in SVMs create a nonlinear decision boundary. In these cases, the RBF kernel used Euclidean distance for flexible nonlinear separation of inhibitors into actives and inactives in Mpro.

			Training Set							Test Set				
Model	Cond.	Split	Non-CV					CV		Non-CV				
			Acc.	Rec.	Prec.	F1	AUC	*k* = 5	*k* = 10	Acc.	Rec.	Prec.	F1	AUC
1	7	0.2	0.78	1.00	0.77	0.87	0.80	0.79	0.79	0.72	0.98	0.72	0.83	0.76
2	8	0.2	0.75	1.00	0.74	0.85	0.86	0.76	0.76	0.72	1.00	0.72	0.84	0.75
3	9	0.2	0.79	1.00	0.78	0.88	0.86	0.79	0.80	0.72	0.98	0.72	0.83	0.80
4	11	0.2	0.78	1.00	0.77	0.87	0.91	0.79	0.78	0.72	1.00	0.72	0.84	0.75
5	13	0.2	0.78	1.00	0.77	0.87	0.91	0.79	0.79	0.73	1.00	0.73	0.84	0.72
6	40	0.2	0.81	0.72	0.76	0.74	0.89	0.82	0.82	0.80	0.70	0.74	0.72	0.85
7	41	0.2	0.83	0.73	0.79	0.76	0.90	0.82	0.83	0.79	0.67	0.73	0.70	0.86
8	42	0.2	0.82	0.73	0.77	0.75	0.90	0.83	0.82	0.78	0.66	0.72	0.69	0.85
9 *	43	0.2	0.84	0.74	0.81	0.78	0.91	0.84	0.84	0.79	0.66	0.75	0.71	0.86
10	45	0.2	0.82	0.70	0.79	0.74	0.89	0.82	0.82	0.77	0.61	0.73	0.66	0.85
11	22	0.25	0.72	0.63	0.70	0.66	0.22	0.72	0.72	0.71	0.63	0.67	0.65	0.23
12	24	0.25	0.73	0.67	0.70	0.69	0.20	0.74	0.73	0.71	0.67	0.66	0.67	0.24
13	24	0.3	0.73	0.64	0.71	0.67	0.20	0.73	0.73	0.72	0.66	0.68	0.67	0.23
14	29	0.2	0.80	0.71	0.80	0.76	0.12	0.80	0.78	0.72	0.63	0.70	0.66	0.22

Abbreviations: acc = accuracy, F1 = F1-score, and AUC = area under curve of the ROC curve. The terms *k* = 5 and *k* = 10 represent the results obtained using 5-fold and 10-fold cross-validation, respectively. RSK (random state of k-fold CV) = 37; RSS (random state for data splitting) = 21; all these SVM models used RBF kernel.

**Table 2 viruses-17-00935-t002:** Performance metrics for different Logistic Regression models. The table shows evaluation metrics for various LR model configurations across training and test sets.

			Training Set							Test Set				
Model	Cond.	Solver	Non-CV					k-Fold CV		Non-CV				
			Acc.	Rec.	Prec.	F1	AUC	*k* = 5	*k* = 10	Acc.	Rec.	Prec.	F1	AUC
1 ^†^	14	liblinear	0.79	0.96	0.79	0.87	0.78	0.78	0.78	0.75	0.97	0.76	0.85	0.71
2 ^†^	14	saga	0.79	0.96	0.79	0.87	0.78	0.78	0.79	0.75	0.97	0.76	0.85	0.70
3 ^†^	14	sag	0.79	0.96	0.79	0.87	0.78	0.78	0.79	0.75	0.97	0.76	0.85	0.70
4 ^†^	14	newton-cg	0.79	0.96	0.79	0.87	0.78	0.78	0.79	0.75	0.97	0.76	0.85	0.70
5 ^†^	14	lbfgs	0.79	0.96	0.79	0.87	0.78	0.78	0.79	0.75	0.97	0.76	0.85	0.70
6	31	lbfgs	0.74	0.66	0.72	0.69	0.80	0.74	0.74	0.71	0.64	0.68	0.66	0.77
7	31	liblinear	0.74	0.66	0.72	0.69	0.80	0.74	0.74	0.72	0.64	0.68	0.66	0.77
8	31	newton-cg	0.74	0.66	0.72	0.69	0.80	0.74	0.74	0.72	0.64	0.68	0.66	0.77
9	31	sag	0.74	0.66	0.72	0.69	0.80	0.74	0.73	0.71	0.64	0.68	0.66	0.76
10 *	45	newton-cg	0.78	0.66	0.73	0.69	0.85	0.78	0.78	0.76	0.60	0.71	0.65	0.83
11	45	sag	0.78	0.65	0.72	0.69	0.85	0.78	0.78	0.76	0.60	0.71	0.65	0.83
12	45	liblinear	0.78	0.66	0.73	0.69	0.85	0.78	0.78	0.76	0.60	0.71	0.65	0.83
13	45	lbfgs	0.78	0.66	0.73	0.69	0.85	0.78	0.78	0.76	0.60	0.71	0.65	0.83

Note: All models employed the RBF kernel. Abbreviations: cond. = condition, rec. = recall, prec. = precision, acc = accuracy, F1 = F1-score, and AUC = area under curve of the ROC curve. The terms *k* = 5 and *k* = 10 represent the results obtained using 5-fold and 10-fold cross-validation, respectively. RSK (random state of k-fold CV) = 11; RSS (random state for data splitting) = 75; ^†^ represents test split of 0.25, otherwise the value represents 0.3.

## Data Availability

The dataset containing the implementation of the machine learning approaches (pipeline and scripts) are available at https://github.com/anacletosouza/Mpro_AI-ML_pipeline/.

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
