# Peer review of "Antagonistic Trends Between Binding Affinity and Drug-Likeness in SARS-CoV-2 Mpro Inhibitors Revealed by Machine Learning"

_viruses, 2025, doi:10.3390/v17070935_

Round 1

Reviewer 1 Report

Comments and Suggestions for Authors

Current manuscript presents an integrated computational pipeline combining machine learning (ML), molecular docking, and molecular dynamics (MD) to explore the trade-off between pharmacodynamic (PD) potency and pharmacokinetic (PK) properties in SARS-CoV-2 main protease (Mpro) inhibitor design. They curated a dataset of ∼55K reported Mpro inhibitors (cleaned to 1,943 compounds with ICâ‚…â‚€ from FRET, fluorescence, and SPR assays) and computed pICâ‚…â‚€ values to label actives/inactives. This work tackles the lack of predictive PK/PD frameworks for Mpro inhibitors, filling a computational niche. However, without experimental or external validation, it only partially bridges this gap—serving more as a proof-of-concept than a definitive solution. In its current form, it reads as a demo study of assembling standard tools on one dataset, rather than a full research report revealing and validating novel mechanisms. Algorithmic components (SVM, LR, docking, MD) are all off-the-shelf, and the study does not experimentally corroborate its mechanistic hypotheses or demonstrate real-world applicability. In addition, authors should involve external or prospective dataset testing, and/or statistical error estimates (confidence intervals, convergence metrics) for the availability of this methodology for real-world application.

Author Response

Attachment

Reviewer 2 Report

Comments and Suggestions for Authors

The goal of this research was to enhance computational screening of anti-SARS-CoV2 drugs versus Mpro (main protease and verified drug target), in part, by combining computational pharmacodynamics (PD) and pharmacokinetic (PK) screens using integrated machine learning (ML), molecular docking and molecular dynamics. A very large number of candidate drugs was screened, and the screen results in the most detailed evaluations of drugs 3AN, 37, 21 and 40, which are Mpro active site inhibitors. The approach is detailed and sophisticated. The paper concludes that subregions of the Mpro active site S2 and S3/S4 are most important for the development of novel anti-Mpro drugs.

More charged drugs tend to improve inhibitor binding affinity (PD) but compromise drug delivery (PK), as expected, but both PD and PK must be considered for each candidate drug.

The paper describes improved methods for computationally pre-screening candidate drugs and evaluating drugs in use. The methodology can also be applied to related viruses and emerging novel threat viruses.

This reviewer has no major concerns about this manuscript. The paper was of high quality, well-written and well-conceived.

The Materials and Methods section is detailed and descriptive.   

Minor points:

“Due to the active site accommodating different subsites, over 55,429 chemical structures have been tested on Mpro experimentally and available in the literature [9]. T” Awkward-rephrase.

“In the case of hepatitis this antivirus is only controlly prescribed. I”—I don’t think “controlly” is a word

“Since the coefficients (provided in the Supplementary File S2) quantify each descriptor’s contribution to predicting” “because” would be better formal English

Author Response

Attachment
